# The Role of Heart Rate Variability (HRV) in Different Hypertensive Syndromes

**DOI:** 10.3390/diagnostics13040785

**Published:** 2023-02-19

**Authors:** Louise Buonalumi Tacito Yugar, Juan Carlos Yugar-Toledo, Nelson Dinamarco, Luis Gustavo Sedenho-Prado, Beatriz Vaz Domingues Moreno, Tatiane de Azevedo Rubio, Andre Fattori, Bruno Rodrigues, Jose Fernando Vilela-Martin, Heitor Moreno

**Affiliations:** 1School of Medical Sciences, State University of Campinas (UNICAMP), Campinas 13083-887, SP, Brazil; 2Post-Graduate Course in Medical Science, Faculty of Medicine of São José do Rio Preto (FAMERP), São José do Rio Preto 15090-000, SP, Brazil; 3Internal Medicine Department, State University of Santa Cruz (UESC), Ilhéus 45662-900, BA, Brazil; 4Cardiovascular Pharmacology & Hypertension Laboratory, School of Medical Sciences, State University of Campinas (UNICAMP), Campinas 13083-887, SP, Brazil; 5Geriatrics Department, School of Medical Sciences, State University of Campinas (UNICAMP), Campinas 13083-887, SP, Brazil; 6Laboratory of Cardiovascular Investigation and Exercise, School of Physical Education, State University of Campinas (UNICAMP), Campinas 13083-887, SP, Brazil

**Keywords:** heart rate variability, prehypertension, hypertension, resistant hypertension, chronic kidney disease, autonomic dysfunction, parasympathetic nervous system, sympathetic nervous system

## Abstract

Cardiac innervation by the parasympathetic nervous system (PNS) and the sympathetic nervous system (SNS) modulates the heart rate (HR) (chronotropic activity) and the contraction of the cardiac muscle (inotropic activity). The peripheral vasculature is controlled only by the SNS, which is responsible for peripheral vascular resistance. This also mediates the baroreceptor reflex (BR), which in turn mediates blood pressure (BP). Hypertension (HTN) and the autonomic nervous system (ANS) are closely related, such that derangements can lead to vasomotor impairments and several comorbidities, including obesity, hypertension, resistant hypertension, and chronic kidney disease. Autonomic dysfunction is also associated with functional and structural changes in target organs (heart, brain, kidneys, and blood vessels), increasing cardiovascular risk. Heart rate variability (HRV) is a method of assessing cardiac autonomic modulation. This tool has been used for clinical evaluation and to address the effect of therapeutic interventions. The present review aims (a) to approach the heart rate (HR) as a CV risk factor in hypertensive patients; (b) to analyze the heart rate variability (HRV) as a “tool” to estimate the individual risk stratum for Pre-HTN (P-HTN), Controlled-HTN (C-HTN), Resistant and Refractory HTN (R-HTN and Rf-HTN, respectively), and hypertensive patients with chronic renal disease (HTN+CKD).

## 1. Introduction

Several studies in animal models of spontaneous hypertension show that alterations in autonomous function are present before the onset of hypertension, suggesting that an increase in sympathetic activity precedes the manifestation of BP elevation, probably associated with a central disturbance of autonomous modulation [1,2,3].

Cardiac innervation by the parasympathetic nervous system (PNS) and the sympathetic nervous system (SNS) acts by modulating the heart rate (HR) (chronotropic activity) and the contraction of the cardiac muscle (inotropic activity). The peripheral vasculature, in turn, is controlled only by the SNS, which is responsible for peripheral vascular resistance. The SNS also mediates the baroreceptor reflex (BR), which mediates blood pressure (BP).

Cardiovascular autonomous dysfunction consists of an imbalance between sympathetic and parasympathetic activity with increased peripheral sympathetic activity and reduced vagal (parasympathetic) tone, and it constitutes an important drive shift of the autonomous function in primary hypertension [4,5,6,7]. Thus, surrogate markers of autonomic regulation investigation, such as HRV, may be useful in the follow-up and evolution of hypertensive syndromes.

This article intends to explore the importance of autonomic modulation assessment through Heart Rate Variability (HRV) as a part of clinical practice in certain pathological circumstances related to HTN. Thus, we aimed: (a) to approach the heart rate (HR) as a CV risk factor; (b) to analyze the heart rate variability (HRV) as a “tool” to estimate the individual risk stratum for Pre-HTN (P-HTN), Controlled-HTN (C-HTN), Resistant and Refractory HTN (R-HTN and Rf-HTN, respectively), and hypertensive patients with chronic kidney disease (HTN+CKD).

## 2. Heart Rate as a Risk Factor

Increases in HR correlate with high BP regardless of gender, age, HTN levels [8,9,10,11], presence of diabetes [12,13,14,15,16], and physical exercises [17,18,19]. High HR is considered a predictor of HTN progression and indicates a need for adjustments in pharmacological therapy [20,21]. Likewise, the ARIC study (Atherosclerosis Risk in Communities Study) has shown that high HR and low HRV were linked with a higher risk of diabetes, obesity, and development of HTN [22]. Additionally, the pre-HTN population with HR > 80 bpm presented increases in all causes of mortality and risk of coronary arterial disease in women [23]. The VALUE study demonstrated, after adjustment for confounders, that the hazard ratio of the composite cardiac primary endpoint (heart failure and total mortality) for a 10-beats/min of the baseline HR increment was 1.16 (95% confidence interval 1.12 to 1.20). Compared to the lowest HR quintile, the adjusted hazard ratio in the highest quintile was 1.73 (95% confidence interval 1.46 to 2.04). Thus, increased HR has been demonstrated as an accurate predictor of long-term cardiovascular events, independently of the achieved BP [6].

Identically, a cohort study (at Glasgow Blood Pressure Clinic) found that high HR is an independent predictor of CV mortality, coronary arterial disease (CAD), and mortality by all causes [24]. For each beat of HR change, there was a 1% change in mortality risk. The highest risk of an all-cause event was associated with patients who had increased their HR by ≥5 bpm at the end of follow-up (1.51 [95% CI: 1.03 to 2.20]; *p* = 0.035). Compared with low–low patients, high–high patients had a 78% increase in the risk of all-cause mortality (HR: 1.78 [95% CI: 1.31 to 2.41]; *p* < 0.001). Therefore, the lowest risk of a CV event was associated with an HR of 61 to 70 bpm. Relative to the baseline HR category (≤60 bpm), patients with an HR of 81 to 90 bpm had the highest risk of all-cause mortality [24].

The results of the SPRINT HR analysis showed that RHR > 80 bpm is associated with a higher hazard ratio for CV outcome in subjects allocated to the intensive treatment arm (hazard ratio, 1.31, with 95% CI, 0.88–1.93) than in subjects allocated to the standard treatment arm (hazard ratio, 1.09, with 95% CI, 0.77–1.52) compared with RHR < 80 bpm [25].

Upwards in HR is also a marker of target organ damage in healthy and HTN-I individuals. The I-SEARCH study [26] showed that subjects with HR between 80–100 bpm had raised microalbuminuria compared to those with HR < 60 bpm [26]/Facila e cols. correlated HR at night >65 bpm with increased prevalence in target-organ damage compared to subjects with HR < 65 bpm [27].

HR increases were associated with accelerated progression in pulse wave velocity (PWV), leading to arterial stiffness, even in controlled HTN patients [28,29,30]. Upward HR oscillations during sleep constitute an independent marker of elevation of both central aortic pressure and aortic augmentation index (Augmentation Index—AIx) [31]. Previous studies relating positive chronotropic effects corroborate a more significant incidence of CV outcomes, target organ damage in HTN, and total mortality [11].

Furthermore, sustained high HR is linked to target organ lesions in HTN, early vascular aging, central hemodynamic disturbances, and CV events [32]. 

Papaioannou et al. [33] demonstrated that increased resting HR is significantly related to higher PWV in subjects with abnormal arterial stiffness (AS), PWV > 10 m/s. In contrast, PWV measurements were not related to resting HR values in subjects with normal AS. These findings were independent of other factors affecting both HR and AS, such as age, sex, BMI (or body height), smoking status, DBP, and the presence of hypercholesterolemia, diabetes, and hypertension [34].

Experimental evidence regarding the influence of HR on arterial stiffness was published by several authors [35,36]. Studies in silico model of the arterial tree observed dependence of PWV on heart rate for PWV values above 9 m/s. Clearly, the magnitude of the heart-rate-dependent change in PWV for low PWV levels is very low, ranging from −0.03 to 0.01 m/s per 5 beats/min. In contrast, for the higher PWV levels, the dependence of PWV on heart rate was stronger [36].

Wilkinson and cols. have demonstrated that 5% higher HR values during cardiac pacing led to a 5.0% fall in the augmentation index (AIx). Nevertheless, a HR higher than 100 bpm did not cause additional decreases in the main parameters [37].

However, the significant effect of HR on PWV, peripheral, and central arterial blood pressures requires further investigation, particularly in older subjects with high-baseline AS or individuals with high BP. The predictive role of PWV in these individuals should be co-evaluated with HR.

## 3. Heart Rate Variability (HRV)

Despite the limitations of measuring the HRV, this parameter is considered an essential and straightforward tool for evaluating autonomic balance in the cardiocirculatory system [38,39,40]. The primary technique is a non-invasive estimative of the balance of variables related to the autonomic innervation (sympathetic–parasympathetic) regulating the duration of the R-R intervals. This paired system, a net of afferent (motor) and efferent (sensoria) nerves, coordinates the basal ganglia, determining a continuous flux of excitatory and inhibitory stimuli, leading to a sympathovagal equilibrium, and therefore stable hemodynamic performance. In physiological conditions, at rest, vagal activity prevails over sympathetic activity in a 4:1 proportion. During physical exercises, this disproportion partially reverts, but even with maximum sympathetic stimulation, some vagal activity stands [39,41].

## 4. Autonomic Nervous System and HRV

The autonomic nervous system (ANS) modulates HR. Activation of the sympathetic nervous system increases HR and decreases HRV, whereas parasympathetic nervous activity decreases HR and increases HRV [42]. The control of the autonomic output involves several interconnected areas of the central nervous system, forming the so-called central autonomic network. In addition to this central control, the arterial baroreceptor reflex, along with respiration, induces quick changes in heart rate. This mechanism influences sympathetic and parasympathetic activity through a specific baroreflex arc.

The normal heart rate variation with breathing is typically the most conspicuous component of the HRV. During inspiration, the vagus nerve stimulation is cut off due to decreased intrathoracic pressure. Consequently, HR increases. During expiration, intrathoracic pressure increases, activating the baroreceptors and vagus nerve stimulation; therefore, HR decreases.

The autonomic system integrity is evaluated by analyzing the HRV. It is assessed by calculating various parameters that describe the magnitude or nature of the interbeat interval variability. 

HRV is the amount by which the time interval between successive heartbeats (interbeat interval, IBI) varies from beat to beat. The magnitude of this variability is small (measured in milliseconds), requiring specialized measurement devices and accurate analysis tools. Typically, HRV is extracted from an electrocardiogram (ECG) measurement by measuring the time intervals between the R-peaks of successive heartbeats, which is considered the gold standard (see Figure 1). However, non-ECG-based methods, such as photoplethysmography (PPG) or speckle plethysmography (SPG), can also be applied. Preferably, such alternative methods should approximate the accuracy of the ECG-derived methods as closely as possible.

HRV in healthy individuals is strongest during resting periods, whereas during stress and physical activity, HRV is decreased. The magnitude of heart rate variability is different between individuals. High HRV is commonly linked to young age, good physical fitness, and good overall health [43,44].

In the clinical scenarios, HRV has been valuable to the risk stratification of sudden cardiac death after acute myocardial infarction [41,45,46,47]. In addition, decreased HRV is generally accepted to provide an early warning sign of diabetic cardiovascular autonomic neuropathy, with the most significant decrease in HRV found within the first 5–10 years of diabetes [48,49,50].

Besides these two main clinical scenarios, HRV has been studied in several cardiovascular diseases, renal failure, physical exercise, occupational and psychosocial stress, gender, age, drugs, alcohol, smoking, sleep, etc. [47,51,52,53,54].

### 4.1. HRV Analysis Methods

HRV analysis should be conducted using long-term ECG signal record (24 h), as well as short-term (ST, ~5 min) or brief and ultra-short term (UST, <5 min) signal record. HRV measurement methods are divided into three main categories: time-domain, frequency-domain, and nonlinear HRV analysis methods.

#### Time-Domain Parameters 

The time-domain HRV analysis methods derive from the beat-to-beat RR. The RR interval time series include successive beat intervals, i.e., (RR = R_1_, R_2_, R_3_… Rn). In some contexts, normal-to-normal (NN) may also be used when referring to these intervals, indicating only intervals between successive QRS complexes resulting from SA-node depolarization.

The most used parameter has two classes: (a) those derived from measurements of the NN intervals or instantaneous HR; and (b) those derived from the differences between NN intervals. The latter method allows the comparison of HRV recorded during daily activities, such as resting and sleeping. Other variables are calculated using specific software. SDNN (Standard Deviation of all NN intervals) represents the standard deviation of all coupling intervals for all consecutive regular beats (NN). SDNN is a global HRV index obtained from continuous electrocardiogram monitoring over 24 h.

Other short-term measures derived from NN interval differences include RMSSD (Root Mean Square of Successive Differences between RR intervals), which represents the average differences between adjacent RR intervals. Physiologically, RMSSD mainly reflects the activity of the parasympathetic nervous system. pNN50 (Percentage of Differences between NN intervals greater than 50 ms) also reflects the parasympathetic nervous system activity and is an excellent marker of failure of autonomic modulation [41,55,56,57,58]. See Table 1.

### 4.2. Analysis of HRV by Geometric Methods

In addition to the above statistical measures, some geometric HRV analysis methods can be calculated based on the RR interval histogram. The HRV triangular index is the integral of the histogram (total number of RR intervals) divided by the height of the histogram, which depends on the selected bin width [55,56,57,58]. Another geometric measure is the TINN, which is the baseline width of the RR histogram evaluated through triangular interpolation (see Figure 2A).

Baevsky’s stress index (SI) [59] is another computed formula in which AMo is the so-called mode amplitude presented in percent, Mo is the mode (the most frequent RR interval), and MxDMn is the variation scope reflecting the degree of RR interval variability. The Mo is simply the median of the RR intervals. The AMo is the height of the normalized RR interval histogram (bin width 50 msec) and MxDMn is the difference between the longest and shortest RR interval values. To make SI less sensitive to slow changes in mean heart rate (which would increase the MxDMn and lower AMo), the very low-frequency trend is removed from the RR interval time series using the smoothness priors method [60]. In addition, the square root of SI is taken to transform the tailed distribution of SI values towards normal distribution (see Figure 2B).

### 4.3. Nonlinear HRV Analysis Methods

Assessment of cardiovascular autonomic reflexes is a critical element in autonomic function evaluation in humans. The complex control systems of the heart with temporal alteration of homeostatic state and this ever-changing dynamicity of ANS output pattern are appropriately reflected in the refined and dynamic alteration of the resting heart rate. HRV results from the dynamic interplay between the multiple physiologic mechanisms regulating the instantaneous HR. Thus, it is reasonable to assume that nonlinear mechanisms are involved in heart rate regulation [47,61,62]. Nonlinear HRV analysis methods include the Poincaré plot [63,64] (a graphical representation of the correlation between RR intervals) and approximate and sample entropy [65,66,67]. Another nonlinear HRV parameter which is used quite often is the detrended fluctuation analysis (DFA), which measures the fractal behavior of HRV [68,69], correlation dimension [70,71], and recurrence plots [72,73,74].

#### Poincare’s Plot

Poincare’s plot is a two-dimensional graphical representation of the correlation between consecutive RR intervals (RRn); each interval is plotted on the *x*-axis versus the next break (RRn+1) on the *y*-axis. It is a quantitative method obtained by adjusting an ellipse to the figure formed by the plot, from which the indices are SD1, SD2, SD1/SD2 [64,75] (see Figure 3). 

SD1 is the instantaneous beat-to-beat variability as a marker of parasympathetic modulation) [76]. SD2 is the variability in long-term continuous RR intervals as parasympathetic and sympathetic modulation markers) [63,77]. SD1/SD2 ratio indicates increased sympathetic modulation during incremental physical exercises [55,78].

This statistical method converts RR intervals into geometric patterns and allows the HRV analysis through the graphic properties resulting in an ellipsoid distribution of points. The analysis can be made qualitatively by evaluating the figure formed by its attractor, which shows the degree of complexity of the RR intervals [61,79].

## 5. Frequency Domain Analysis (Spectral Density)

The analysis in the frequency domain (spectral density) describes the oscillations of the heart rate signal decomposed at different frequencies and amplitudes. It provides information on the number of its relative intensities in the sinus heart rate. There are two ways to perform spectral analysis [47,55]. The first is the Fast Fourier Transformation (FFT), a nonparametric method which records spectral peaks for the various components of the frequency [80]. The second is the autoregressive model, a parametric method that registers a continuous spectrum of activities.

The first method is faster, since the autoregressive model is more complex and needs a suitability check. When using the FFT, the computer transforms the individual RR intervals stored into spectral bands of different frequencies (see Figure 4).

The duration of the RR intervals obtained in milliseconds is converted into heart rate per minute. The unit used is Hertz (cycles per second). The HRV total amplitude of the HRV spectrum consists of four bands: the ultra-low frequency (ULF < 0.003 Hz), the very-low-frequency (VLF 0.003–0.04 Hz), the low-frequency (LF 0.04–0.15 Hz), and the high frequency (HF 0.15–0.5 Hz) bands. The VLF component is related to fluctuations in the vasomotor tone involved in thermoregulation and sweating (sympathetic control). 

The LF component is associated with baroreceptors (sympathetic control on the parasympathetic modulation) and mainly measures the sympathetic tone [81,82]. The HF component is related to parasympathetic activity [83,84,85,86]. Finally, the LF/HF ratio reflects the sympathetic over parasympathetic activity. Figure 5 summarizes the most used frequency domain parameters. Short spectral records (5–10 min) are characterized by the components VLF, HF, and LF, while long-term recordings (24 h) include all the previous components along with the ULF [55].

The very-low-frequency (VLF) band of HRV has distinct characteristics compared to other components of HRV [87,88]. Two main mechanisms have been proposed to explain this difference. The first is related to the modulation of thermoregulation, which increases the frequency of VLF, which is reflected in heart rate variability [89]. In this sense, changes in external temperature can modify cutaneous blood flow and RR interval. However, there is no evidence that cutaneous thermoregulation, RR intervals, and blood pressure fluctuate simultaneously. 

The second proposed mechanism demonstrates the influence of the renin–angiotensin–aldosterone system on VLF band recording [90]. In experimental work, Akselrod et al. demonstrated that ACE blockade increases VLF spectral power [87]. Clinical studies using 24 h ECG monitoring have confirmed this association. Moreover, recordings obtained from post-myocardial infarction patients [91] and patients with congestive heart failure [92] showed that ACE blockade increases HRV in the time domain and VLF.

In addition, the VLF band has a strong association with coronary heart disease, [91,93] stroke [94,95], and metabolic syndromes [96]. Furthermore, high VLF power is associated with a high exercise capacity [97]. However, the mechanisms of these VLF associations remain unclear.

HRV assessment has some limitations. Depending on the algorithms used, the number of variables extracted can vary and may include redundant variables, which can complicate the interpretation of the results and decrease the importance of the method. The interpretation of HRV indexes varies and depends on the specific context (including rest, posture, stress, and medications) and individual characteristics, such as age, gender, and the presence of diseases such as diabetes or coronary artery disease. 

In this context, advanced statistical analysis methods that combine non-parametric, robust, and resampling techniques can be practical tools (for example, graphical analysis) for more accessible clinical applications and may identify unexpected relationships between variables [98]. Among these advanced statistical tools, a unitary Autonomic Nervous System Index (ANSI) can help reveal new aspects of the relationship between baroreflex and autonomic indexes by providing valuable analyses of the trends examined [55,99].

## 6. HRV and Hypertensive Syndromes

### 6.1. Heart Rate Variability in Prehypertension

Several reports have shown a significant association between autonomic dysfunction and prehypertension [7,100,101]. The Framingham Heart Study showed that lower HRV was associated with a greater risk of developing hypertension in normotensive men. Additionally, LF spectral analysis improves the prediction of risk of hypertension in men in a 4-year cumulative incidence of hypertension [7]. 

Lucini et al. [102] demonstrated that subjects with blood pressure values in the upper normotensive range present alterations in HRV parameters, which were particularly apparent in the youngest group, suggesting an autonomic shift toward sympathetic predominance and vagal withdrawal [98,102].

Prakash et al. [103] compared the autonomic function using BP, HR, indices of short-term HRV during supine rest and quiet standing, HR variation during timed deep breathing (HRVdb) and pressor responses to the cold pressor test and sustained isometric handgrip in different three groups. The first group included subjects with recent-onset hypertension that were not undergoing treatment. The second group consisted of subjects with high–normal blood pressure (BP). The third was composed of subjects with normal BP. Although the three groups were comparable (*p* > 0.1) in terms of mean HR and low-frequency (LF) power expressed in normalized units at rest and during quiet standing, the standard deviation of normal-to-normal RR intervals (SDNN) during supine rest, LF, and high-frequency spectral powers during supine rest and HRVdb were lowest in hypertensives (*p ≤* 0.05 for each), indicating diminished baroreflex modulation of RR intervals in hypertensives. In contrast, LF power was highest in subjects with high–normal BP (*p ≤* 0.05) during supine rest, and higher BP variability is a possible reason for this.

The outcomes suggest that HRVdb provides a simple measure of cardiac vagal effects in hypertensives. The rate–pressure product provides a simple measurement of overall HRV in hypertensives. In clinical hypertension, the arterial baroreflex mechanism resets to maintain a higher BP through diminished vagal modulation of HR and possibly heightened sympathetic outflow to the heart and resistance vessels [103].

Recently, Hoshi et al.’s [100] results reinforced the hypothesis that autonomic dysfunction is present in the early stage of hypertension and precedes hypertension onset early in the clinical course of high blood pressure (BP) development, playing an important role in its pathogenesis [100].

There is some evidence that prehypertensive individuals already show autonomic impairment compared to subjects with normal BP. An early disturbance of oscillatory properties of HRV may represent a liability for the development of overt arterial hypertension [101].

### 6.2. Heart Rate Variability and Hypertension

An impaired autonomic nervous function has been implicated in the development of coronary heart disease [4,5,101,104,105,106], all-cause mortality [107], and hypertension [4,98,108]. The Atherosclerosis Risk in Communities (ARIC) study found that the hazard ratio for hypertension development during the 9-year follow-up period was 1.36 times higher for participants in the lowest HRV quartile of the square root of the mean-squared differences (rMSSD) compared with those in the highest quartile. In a 4-year Framingham Heart cohort study, low-frequency (LF) power was associated with hypertension incidence during the 4-year follow-up period, with an odds ratio of 1.38 per 1 s.d. of LF decrement. Therefore, impaired autonomic nervous function and decreased HRV may be underlying causes of hypertension, and estimation of HRV may improve hypertension risk prediction [109].

Mori H et al. [110] found significant associations between decreased HRV levels and increased BP regardless of several confounders using the 5 min R-R interval measurement method. Additionally, the study reported a significant inverse association of HRV with diastolic blood pressure. These results are consistent with previous findings for Caucasian populations and confirmed the inverse association between HRV and blood pressure in the Asian population, in which individuals have much lower mean BMIs than those in the United States and Europe [111].

### 6.3. Heart Rate Variability and Resistant Hypertension

Resistant hypertension (RHTN) is characterized by uncontrolled blood pressure (BP) values despite the concomitant use of at least three antihypertensive drugs, one of which is a diuretic [112,113].

Autonomic imbalance, characterized by a hyperactive sympathetic system and a hypoactive parasympathetic system, is associated with various pathological conditions. Nevertheless, it has not been formally addressed in RHTN patients. Measures of HRV in both time and frequency domains have been used successfully to index vagal activity [114]. Nevertheless, while there are some differences among HRV parameters found in many studies, the consensus is that lower values of these indices of vagal function are associated prospectively with death and disability [46].

Parasympathetic activity and HRV have been associated with immune dysfunction and inflammation, which correlate with many conditions, including CVD and diabetes. The association between HRV and RHTN in Type 2 Diabetes Mellitus (T2DM) and non-T2DM was reported by Martins et al. [115] HRV was used to assess autonomic imbalance (AI). Both groups showed disruption of the circadian rhythm, inverted sympathetic and parasympathetic tones (during the day parasympathetic tone > sympathetic and, at night-time periods, sympathetic > parasympathetic tone). HRV in the time domain revealed greater AI in T2DM RHTN compared to non-T2DM RHTN. 

Salles et al. [114] reported a reduced HRV associated with a blunted nocturnal blood pressure fall (non-dipping) in RHTN patients. The HRV parameters analyzed were the SDNN and SDANN, which mainly reflect SNS overactivity in non-dipping RHTN patients.

De La Sierra et al. [116] conducted a study comparing HR in controlled HTN versus uncontrolled RHTN patients. They performed ambulatory BP monitoring (ABPM) and HR monitoring without analyzing HRV in the time and frequency domains. The study showed an association between increased HR and blunted nocturnal HR values in RHTN. This association was more evident in uncontrolled RHTN versus controlled HTN. The significance of these results corroborates the importance of the association between high sympathetic activity and resistance to pharmacological therapies. 

Recently, Rubio et al. demonstrated that uncontrolled RHTN patients display negative changes in autonomic balance compared with controlled RHTN. These results reinforce the importance of autonomic nervous system interventions in managing arterial hypertension [117].

These studies open a window of options for better understanding the possible mechanisms of new therapeutic strategies, such as renal denervation or carotid stimulation, developed for the treatment of RHTN [118].

### 6.4. Heart Rate Variability and Chronic Renal Disease

Chronic kidney disease (CKD) is present in 13.4% of the adult population. It is more prevalent in women than men [119]. In individuals over 65 years of age, even in the absence of diabetes and hypertension, CKD prevalence is greater than 12% [120]. A potential little-explored cause of kidney injury is an autonomic imbalance (high sympathetic tone and/or low parasympathetic tone) [121,122]. Experimental studies demonstrated that changes in autonomic activity modulate renal hemodynamics, tubular transport, and renin secretion [123,124,125]. In humans, several phenomena related to autonomic imbalance, such as impaired diurnal BP variation, have been associated with chronic kidney disease and its progression, regardless of mean arterial pressure and diabetes [126].

Some cross-sectional studies have shown that a lower HRV was significantly associated with traditional and nontraditional risk factors for CVD, including diabetes, lower albumin, and higher phosphorus, CRP, and eGFR < 15 mL/min/1.73 m^2^. Lower HRV was independently associated with a higher risk of CV events, composite of CVD/death, and ESRD [127].

The PREVEND Study [128] included 4605 subjects (49% males, age range = 33–80, 0.6% blacks) and 341 new CKD patients during a median follow-up duration of 7.4 years. Low SDNN (HRV) was associated with a greater incidence of CKD (crude HR = 1.66, 95%. and CI = 1.30 to 2.12, *p* < 0.001], but this finding did not remain significant after adjustment for age, gender, and CV and known risk factors (adjusted HR = 1.13, 95% CI = 0.86 to 1.48, *p* = 0.40). An analysis of renal function over time in the total sample revealed no evidence of a progressive decline in eGFR or an increase in UAE in those with low HRV. These results suggest that reduced HRV may be a complication of CKD rather than a causal factor. In patients with stage 5 CKD, the proportion of abnormal LF, HF, and LF/HF ratio were 69.5, 52.8, and 50%, respectively. The factors related to low HRV included advanced CKD, diabetes mellitus, low serum albumin, and proteinuria. Multivariate logistic regression analysis revealed lower LF/HF, hypertension, and severe proteinuria as faster CKD progression risk factors [128].

Recent publications have shown that repeated heart rate variability (HRV) measurements are more relevant than a single HRV measurement as a predictor of long-term mortality in chronic hemodialysis patients. The authors found that increased nHF values (hazard ratio [HR] 1.033, 95% confidence interval [CI] 1.029–1.036, *p* < 0.001) during the hemodialysis process were an independent predictor for 8-year cardiovascular mortality, although time-domain HRV indices (such as the SDNN) were not used. Meanwhile, increased levels of VLF (HR 0.990, 95% CI 0.986–0.993, *p* < 0.001), Variance (HR 0.991, 95% CI 0.987–0.994, *p* < 0.001), nLF (HR 0.999, 95% CI 0.999–1.000, *p* = 0.006), and LF/HF ratio (HR 0.796, 95% CI 0.746–0.849, *p* < 0.001) during the hemodialysis process protected the patients from subsequent cardiovascular mortality [129].

Thus, patients with CKD have autonomic dysfunction demonstrated by methods of HRV assessment. Moreover, there is evidence of baroreceptor sensitivity deficiency in chronic renal failure patients and changes related to increased cardiovascular morbidity and mortality [130,131].

Reduced HRV is an important marker of cardiovascular risk global mortality and progression of kidney injury in patients with CKD [132]. In this group of patients, this ANS assessment tool has the potential to guide the choice of therapeutic classes that act on ANS [133].

The therapeutic target of CKD patients includes direct and indirect nephroprotection (BP control) and provides favorable effects for the ANS. Concerning parasympathetic changes, there is evidence that some drugs can improve vagal control of heart rate, as assessed by spectral analysis of HRV. This is the case with beta-blockers, angiotensin II receptor antagonists, and, although not always homogeneously, angiotensin-converting enzyme (ACE) inhibitors.

The sympathetic hyperactivity typically presented by patients with CKD can be favorably affected by drugs that act on the renin–angiotensin system, such as ACE inhibitors and angiotensin II receptor blockers.

The mechanisms responsible for the sympathomodulatory properties are (1) a reduction in the excitatory effects of angiotensin II on the peripheral and central adrenergic neural drive, (2) partial or complete restoration of the sympathoinhibitory properties exerted by the baroreflex, and (3) a direct effect of central sympathoinhibitory drugs [134].

Figure 6—The illustrative figure shows the results of the autonomic function in a normal subject.

Figure 7—An illustrative figure of the autonomic dysfunction in a patient with resistant hypertension. 

## 7. Conclusions

Heart Rate Variability (HRV), the beat-to-beat variation in HR, or the duration of the RR-interval, is a straightforward tool for clinical research in Hypertensive Syndromes. 

The balance of the sympathetic and parasympathetic nervous activity plays an important role in HRV, as decreased HRV is a marker of an imbalance of the autonomic system. 

The reduced HRV in prehypertensive individuals is an early manifestation of autonomous dysfunction that may predict the development of clinical hypertension. In hypertensive individuals, modulating ANS activity may allow adequate blood pressure control with increased cardiovascular protection.

In patients with CKD, the assessment of sympathetic–parasympathetic balance has the potential to guide the choice of therapeutic classes that act on the ANS and allow nephroprotection and control of BP.

In resistant hypertensives, despite drug interference in the assessment of ANS, the reduction of HRV as a manifestation of autonomic dysfunction helps to identify patients associated with a better outcome after renal denervation.

Finally, reduced HRV has the potential to provide optimal treatment for different hypertensive syndromes.

## Figures and Tables

**Figure 1 diagnostics-13-00785-f001:**
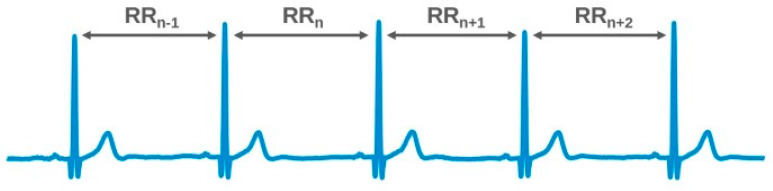
RR interval measurements on an ECG.

**Figure 2 diagnostics-13-00785-f002:**
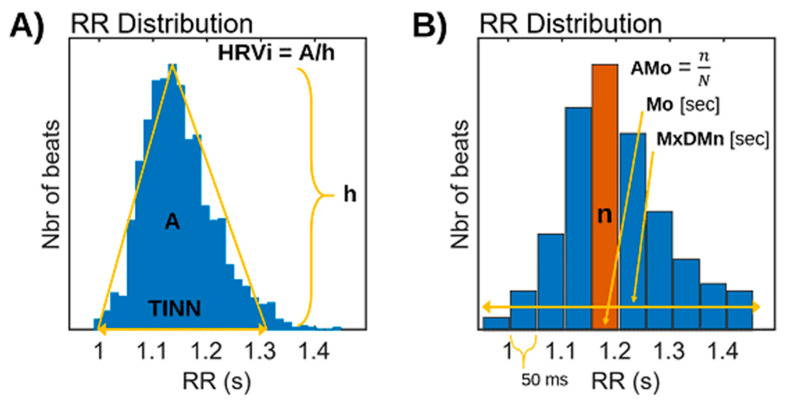
(**A**) Analysis of HRV by geometric methods. Triangular Indexes of Heart Rate Variability (RRtri and TINN). h: height of the yellow triangle. (**B**) Baevsky’s stress index (SI).

**Figure 3 diagnostics-13-00785-f003:**
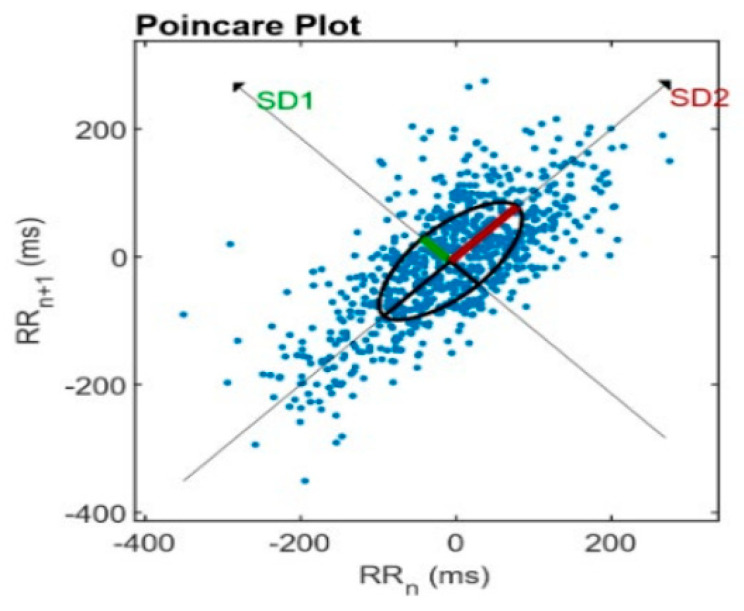
Analysis of the Poincare’s plot—Consecutive RR intervals (RRn) are plotted on the *x*-axis versus the next interval (RRn+1) on the *y*-axis.

**Figure 4 diagnostics-13-00785-f004:**
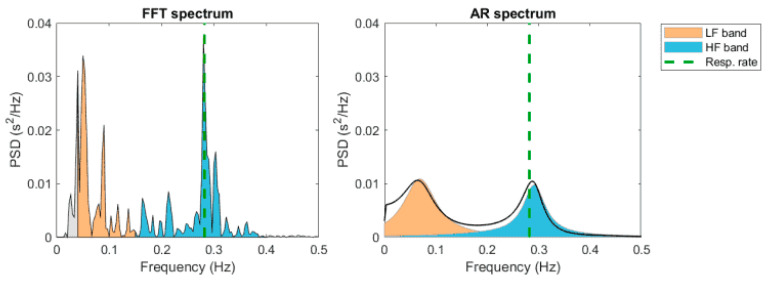
HRV spectrum estimates using FFT-based Welch’s periodogram method (left) and autoregressive (AR) modeling-based spectrum estimation method.

**Figure 5 diagnostics-13-00785-f005:**
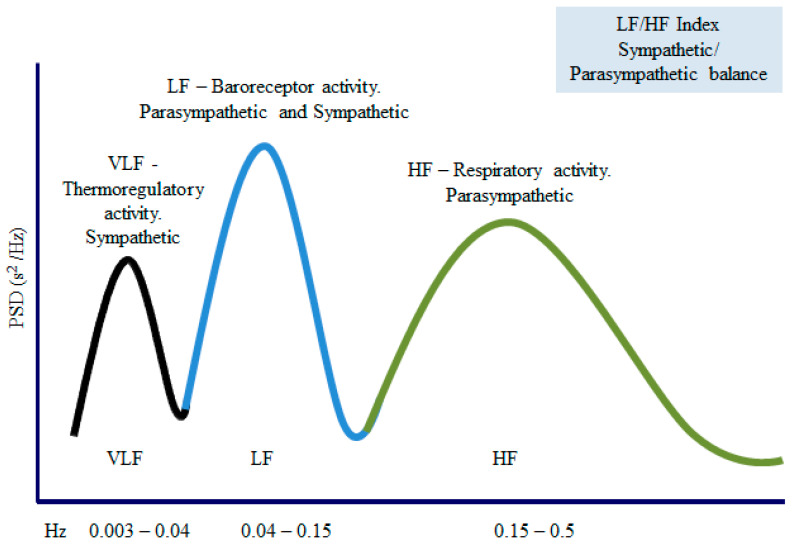
Frequency (Hz) and amplitude (ms^2^) evaluate the spectral components. The area under each part allows the division of the spectral density into bands of frequencies. VLF: very-low-frequency band; LF: low-frequency band; HF—high-frequency band.

**Figure 6 diagnostics-13-00785-f006:**
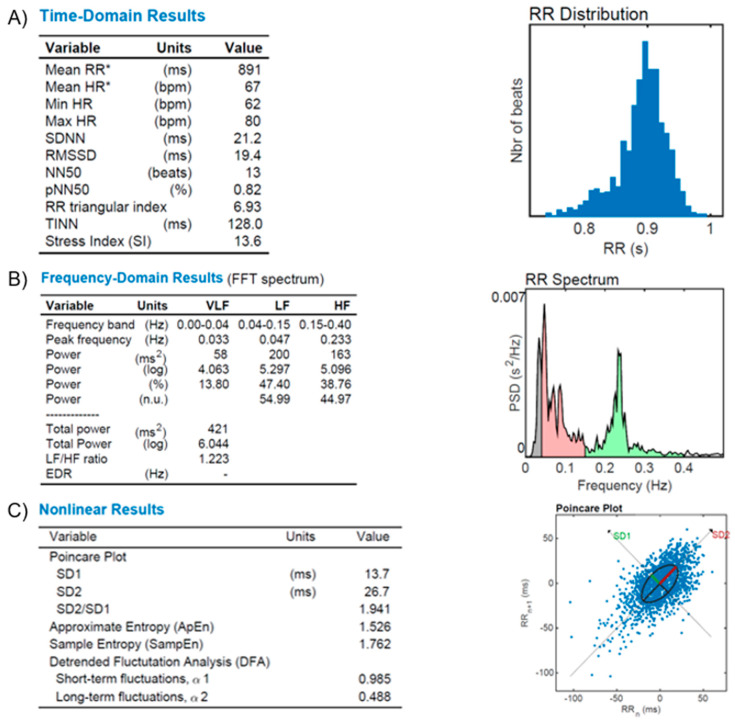
HR Variability (HRV) reports from a healthy subject. (**A**): Time-domain variables analysis (Mean RR, Mean HR, Min HR, Max HR, SDNN, RMSSD, NN50, pNN50, RR triangular index, and TINN, as described in the text); (**B**): frequency-domain variables analysis (FFT spectrum): Very Low (VLF), Low (LF) and High (HF) bands of frequencies; (**C**): Nonlinear results analysis with Poincare’s plot based on RR distribution and geometrical modeling for an ellipsis. Figures from our archive made with the software Kubios. *: Results are calculated from the non-detrended selected RR series.

**Figure 7 diagnostics-13-00785-f007:**
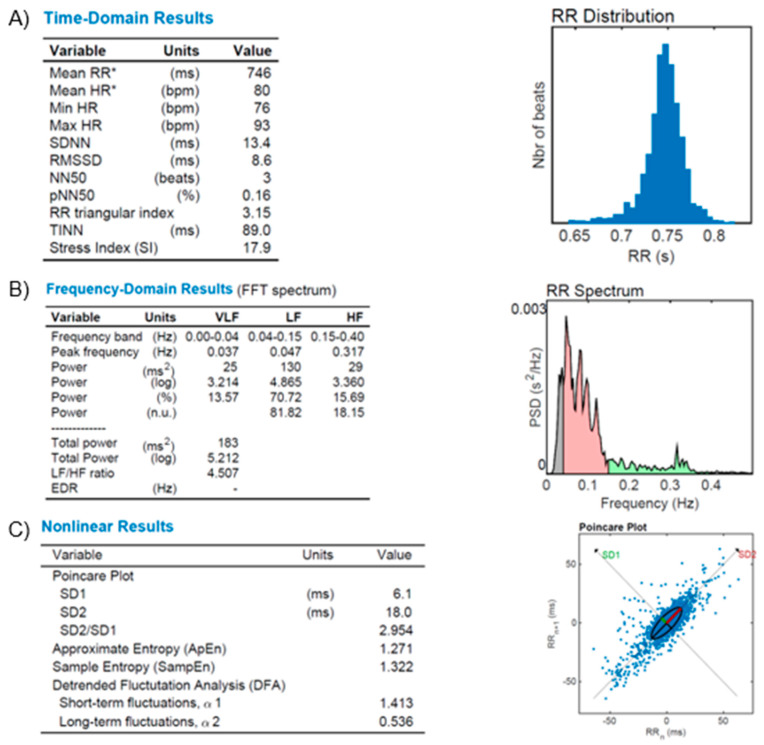
HR Variability (HRV) reports from a diabetic and resistant hypertensive subject. (**A**): time-domain variables analysis (Mean RR, Mean HR, Min HR, Max HR, SDNN, RMSSD, NN50, pNN50, RR triangular index, and TINN, as described in the text); (**B**): frequency-domain variables analysis (FFT spectrum): Very Low (VLF), Low (LF) and High (HF) bands of frequencies; (**C**): Nonlinear results analysis with Poincare’s Plot based on RR distribution and geometrical modeling for an ellipsis. Figures from our archive made with the software Kubios. *: Results are calculated from the non-detrended selected RR series.

**Table 1 diagnostics-13-00785-t001:** HRV Time-domain measures.

Parameter	Unit	Description
HR	bpm	Heart rate
HR max–HR min	bpm	Average difference between the highest and lowest heart rates during each respiratory cycle
SDNN	ms	Standard deviation of NN intervals
RMSSD	ms	Root mean square of successive RR interval differences
pNN50	%	Percentage of successive RR intervals that differ by more than 50 ms

## Data Availability

No new data were created or analyzed in this study. Data sharing is not applicable to this article.

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
