# Peer review of "The Role of Heart Rate Variability (HRV) in Different Hypertensive Syndromes"

_diagnostics, 2023, doi:10.3390/diagnostics13040785_

Round 1

Reviewer 1 Report

A very nice manuscript, well-written and pleasant to read.

Whereas the manuscript rightfully so pronounces that longitudinal measurements over longer duration are preferred, it does not mention non-ECG based methods to monitor HR and HRV, whereas methods like PPG are presently frequently used in wearable devices designed for long duration HR and HRV monitoring. Using PPG as a proxy for ECG RR intervals of course decreases accuracy, but recently a interesting comparison between PPG and SPG (speckle plethysmography) revealed that the latter has a much better accuracy match with ECG RR intervals than PPG. Of course this manuscript does not form a review of technlogies on HOW o measure HR and HRV, but more focuses on you can DO with such measurements. It might, however be worthwhile to somewhere mention that the value of using differences of HR and HRV for diagnostic purposes is directly influenced by the accuracy of the method to measure the individual heart beats. In lines 156-157 we read now: "Typically, HRV is extracted from an electrocardiogram (ECG) measurement by measuring the time intervals between successive heartbeats as illustrated in figure 1." One might modify to something like: "Typically, HRV is extracted from an electrocardiogram (ECG) measurement by measuring the time intervals between the R-peaks of successive heartbeats, which is considered the gold standard (see figure 1). But also non-ECG based methods, like photoplethysmography (PPG) or speckle plethysmograhy (SPG) can be applied. Preferably such alternative methods should approximate the accuracy of the ECG-derived methods as close as possible."

Line 57: "This review intends to explore". Is this manuscript a review? To a certain extent it is, but that would be closer to a narrative review then a systematic review (no criteria in the methods, etc). Suggest to reword to either "this article" (preferred) or at least tune down to "this narrative review".

Line 276, add "be": can be practical tools

In Line 307, the word "is" seems to be an unintended remainder of corrections, as it is printed in strikethrough font.

Lines 332-335: Suggest to strengthen the connection between 2 sentences  "These results are consistent with previous findings for Caucasian populations and confirmed the inverse association between HRV and blood pressure in the Asian population, where individuals have much lower mean BMIs than those in the United States and Europe". I think this is more what you intended to state.

Figures 6 and 7 seem added as examples to illustrate typical differences between a healthy person versus someone being diabetic and resistant to treatment for hypertension. In the RR distribution histogram of figure 6, the x-axis label values are printed confusingly close to each other. When comparing with the x-axis used in figure 7, the ranges differ a lot. Suggest to use the same x-axis range in both figures. Upon a first glance on the figures, one gets the impression of markedly different poincare plots. But when looking at x and y axis scale ranges, one realizes these differences are more subtle than they look. Suggest to use the same x and y axis ranges in both figures. There is something weird with the headings of the frequency domain tables in both figures, please increase readability.

Author Response

A very nice manuscript, well-written and pleasant to read.

R: Dear reviewer, thanks for your excellent suggestions about the paper. The response of the suggested changes is typed in blue. 

Whereas the manuscript rightfully so pronounces that longitudinal measurements over longer duration are preferred, it does not mention non-ECG based methods to monitor HR and HRV, whereas methods like PPG are presently frequently used in wearable devices designed for long duration HR and HRV monitoring. Using PPG as a proxy for ECG RR intervals of course decreases accuracy, but recently a interesting comparison between PPG and SPG (speckle plethysmography) revealed that the latter has a much better accuracy match with ECG RR intervals than PPG. Of course this manuscript does not form a review of technlogies on HOW o measure HR and HRV, but more focuses on you can DO with such measurements. It might, however be worthwhile to somewhere mention that the value of using differences of HR and HRV for diagnostic purposes is directly influenced by the accuracy of the method to measure the individual heart beats. In lines 156-157 we read now: "Typically, HRV is extracted from an electrocardiogram (ECG) measurement by measuring the time intervals between successive heartbeats as illustrated in figure 1." One might modify to something like: "Typically, HRV is extracted from an electrocardiogram (ECG) measurement by measuring the time intervals between the R-peaks of successive heartbeats, which is considered the gold standard (see figure 1). But also non-ECG based methods, like photoplethysmography (PPG) or speckle plethysmograhy (SPG) can be applied. Preferably such alternative methods should approximate the accuracy of the ECG-derived methods as close as possible."

R: We agree with the reviewer, the text was modified in the original manuscript following the suggestion of the reviewer.

Line 57: "This review intends to explore". Is this manuscript a review? To a certain extent it is, but that would be closer to a narrative review then a systematic review (no criteria in the methods, etc). Suggest to reword to either "this article" (preferred) or at least tune down to "this narrative review".

R: We agree with the reviewer and the appropriate rewrite was made.

Line 276, add "be": can be practical tools

R: The word correction was made.

In Line 307, the word "is" seems to be an unintended remainder of corrections, as it is printed in strikethrough font.

R: The incorrect word was deleted.

Lines 332-335: Suggest to strengthen the connection between 2 sentences  "These results are consistent with previous findings for Caucasian populations and confirmed the inverse association between HRV and blood pressure in the Asian population, where individuals have much lower mean BMIs than those in the United States and Europe". I think this is more what you intended to state.

R: We agree with the reviewer. The intention was exactly that.

Figures 6 and 7 seem added as examples to illustrate typical differences between a healthy person versus someone being diabetic and resistant to treatment for hypertension. In the RR distribution histogram of figure 6, the x-axis label values are printed confusingly close to each other. When comparing with the x-axis used in figure 7, the ranges differ a lot. Suggest to use the same x-axis range in both figures. Upon a first glance on the figures, one gets the impression of markedly different poincare plots. But when looking at x and y axis scale ranges, one realizes these differences are more subtle than they look. Suggest to use the same x and y axis ranges in both figures. There is something weird with the headings of the frequency domain tables in both figures, please increase readability.

R: We agree with the reviewer. The necessary corrections were made in the figures 6 and 7.

Reviewer 2 Report

This article reviews the scientific evidence for the value of heart rate variability in predicting complications of arterial hypertension.

Because there have been too many publications on the relationship between hypertension and heart rate variability over the last 40-50 years, the title of the review does not reflect originality or novelty: Heart Rate Variability (HRV) in Hypertensive Syndromes: How and Why?

Consequently, the title should reflect the specific nature of the pathology being studied and the role of heart rate variability in the diagnosis of this pathology.

The authors focus their review on special forms of arterial hypertension - prehypertension, therapy-resistant hypertension and hypertension with impaired renal function.A review of information reflecting the relationship between heart rate and arterial hypertension was presented in sufficient detail. Unfortunately, there is not enough information on the physiological and clinical interpretation of the ultra-low-frequency part of heart rate variability (VLF) and non-linear parameters of heart rate variability, which reflect hemodynamic processes. How useful are these parameters in assessing exactly resistant arterial hypertension, pre-hypertension and hypertension in the presence of kidney pathology?

Therefore, the Conclusion should contain prospects for using the method of heart rate variability in these forms of arterial hypertension (resistant arterial hypertension, pre-hypertension and hypertension in the presence of kidney pathology).

Author Response

This article reviews the scientific evidence for the value of heart rate variability in predicting complications of arterial hypertension.

R: Dear reviewer, thanks for your excellent suggestions about the paper. The response of the suggested changes is typed in blue. 

Because there have been too many publications on the relationship between hypertension and heart rate variability over the last 40-50 years, the title of the review does not reflect originality or novelty: Heart Rate Variability (HRV) in Hypertensive Syndromes: How and Why?

R: The title was modified.

Consequently, the title should reflect the specific nature of the pathology being studied and the role of heart rate variability in the diagnosis of this pathology.

R:  We agree with the reviewer. The title was changed to: “The role of Heart Rate Variability (HRV) in different Hypertensive Syndromes”; Running title: Heart Rate Variability in Hypertensive Syndromes.

The authors focus their review on special forms of arterial hypertension - prehypertension, therapy-resistant hypertension and hypertension with impaired renal function. A review of information reflecting the relationship between heart rate and arterial hypertension was presented in sufficient detail. Unfortunately, there is not enough information on the physiological and clinical interpretation of the ultra-low-frequency part of heart rate variability (VLF) and non-linear parameters of heart rate variability, which reflect hemodynamic processes. How useful are these parameters in assessing exactly resistant arterial hypertension, pre-hypertension and hypertension in the presence of kidney pathology?

 R: We agree with the reviewer.

The new paragraph about the VLF analysis was inputed in the text.

The nonlinear parameters were rewritten in the manuscript text.

Therefore, the Conclusion should contain prospects for using the method of heart rate variability in these forms of arterial hypertension (resistant arterial hypertension, pre-hypertension, and hypertension in the presence of kidney pathology).

R: We agree with the reviewer. The conclusion was meant to emphasise the importance of the HRV evaluation in the different forms of arterial hypertension: prehypertension, resistant hypertension, and hypertension in presence of chronic kidney disease.